# Development of Nonstructural Protein-Based Indirect ELISA to Identify Elephant Endotheliotropic Herpesvirus (EEHV) Infection in Asian Elephants (*Elephas maximus*)

**DOI:** 10.3390/ani12141747

**Published:** 2022-07-07

**Authors:** Thunyamas Guntawang, Tidaratt Sittisak, Pallop Tankaew, Chatchote Thitaram, Varangkana Langkapin, Taweepoke Angkawanish, Tawatchai Singhla, Nattawooti Sthitmatee, Wei-Li Hsu, Roongroje Thanawongnuwech, Kidsadagon Pringproa

**Affiliations:** 1Department of Veterinary Biosciences and Veterinary Public Health, Faculty of Veterinary Medicine, Chiang Mai University, Chiang Mai 50100, Thailand; thunyamas.g@gmail.com (T.G.); orilsai_o_s@hotmail.com (T.S.); drneaw@gmail.com (N.S.); 2Veterinary Diagnostic Center, Faculty of Veterinary Medicine, Chiang Mai University, Chiang Mai 50100, Thailand; pallop_off@hotmail.com; 3Center of Elephant and Wildlife Health, Chiang Mai University, Chiang Mai 50100, Thailand; cthitaram@gmail.com; 4Department of Companion Animals and Wildlife Clinics, Faculty of Veterinary Medicine, Chiang Mai University, Chiang Mai 50100, Thailand; 5National Elephant Institute, Forest Industry Organization, Lampang 52190, Thailand; w.langkaphin@gmail.com (V.L.); taweepoke@gmail.com (T.A.); 6Department of Food Animal Clinic, Faculty of Veterinary Medicine, Chiang Mai University, Chiang Mai 50100, Thailand; tawatchai.singh@cmu.ac.th; 7Graduate Institute of Microbiology and Public Health, College of Veterinary Medicine, National Chung Hsing University, Taichung 402, Taiwan; wlhsu@dragon.nchu.edu.tw; 8Department of Pathology, Faculty of Veterinary Sciences, Chulalongkorn University, Bangkok 10330, Thailand; roongroje.t@chula.ac.th

**Keywords:** EEHV, indirect ELISA, antibody, nonstructural protein, active infection

## Abstract

**Simple Summary:**

Elephant endotheliotropic herpesvirus hemorrhagic disease (EEHV-HD) is one of the most important viral infectious diseases in young Asian elephants (*Elephas maximus*). To date, adult elephants that have been infected with EEHV have displayed mostly mild clinical signs, while they are believed to serve as EEHV shedders to other elephants. Hence, the diagnostic tools employed for monitoring EEHV-active infection are important. In this study, partial EEHV-DNA polymerase (DNApol) nonstructural proteins (NSPs) were used to detect anti-EEHV antibodies through the use of an in-house indirect enzyme-linked immunosorbent assay (ELISA), in comparison to the results of a PCR test. The developed EEHV-DNApol ELISAs demonstrated values of 77.9% sensitivity and 87.7% specificity, respectively. The results demonstrated that the sera obtained from elephants that exhibited no clinical signs of EEHV infection, and who were negative according to PCR tests of the blood, reported values of 14% seropositivity for EEHV-DNApol. Our results indicate that these asymptomatic, active EEHV-infected elephants could likely serve as the source of EEHV shedding in the herd. The developed EEHV-DNApol NSPs-based ELISA test employed in the present study may be of use for routine monitoring and identification of EEHV shedders in elephant herds, and could be an alternative diagnostic tool for EEHV detection in Asian elephants.

**Abstract:**

Disease caused by elephant endotheliotropic herpesvirus (EEHV) infection is the most highly fatal hemorrhagic disease in Asian elephant calves worldwide. To date, adult elephants that have been infected with EEHV have predominantly displayed mild clinical signs, while they are believed to serve as EEHV shedders to other elephants. Hence, the diagnostic tools employed for monitoring EEHV-active infection are considered vitally important. In this study, partial EEHV-DNA polymerase (DNApol) nonstructural proteins (NSPs) were used to detect anti-EEHV antibodies through the use of an in-house indirect enzyme-linked immunosorbent assay (ELISA). The results were then compared to those obtained from a PCR test. In this study, a total of 175 serum samples were collected from Asian elephants living in elephant camps located in Chiang Mai and Lampang Provinces, Thailand. The elephants were aged between 2 and 80 years old. The overall percentages of positive samples by the PCR and EEHV-DNApol ELISA tests were 4% (21/175) and 12% (21/175), respectively. The ELISAs demonstrated values of 77.9% (95% posterior probability interval (PPI) = 52.5–95%) sensitivity and 87.7% (PPI = 82.5–91.9%) specificity, respectively. Accordingly, the sera obtained from the elephants exhibiting no clinical signs of EEHV infection, and those who were negative according to PCR tests, revealed a value of 14% seropositivity for EEHV-DNApol. Our results indicate that these asymptomatic, active EEHV-infected elephants could likely serve as a source of EEHV shedding within elephant herds. Consequently, the developed EEHV-DNApol NSPs-based ELISA test employed in the present study may be of use for routine monitoring and identification of EEHV shedders in elephant herds, and could be an alternative diagnostic tool for EEHV detection in Asian elephants.

## 1. Introduction

Elephant endotheliotropic herpesvirus (EEHV) is one of the most virulent and highly fatal viral infections affecting Asian elephants (*Elephas maximus*) worldwide [1,2]. EEHV is an enveloped, double-stranded DNA virus that belongs to the species *Elephantid betaherpesvirus*, subfamily *Betaherpesvirinae* and genus *Proboscivirus* [3]. Most EEHV-infected cases are associated with lesions, edema, and hemorrhage in internal organs [2,3,4]. Based on genetic distinctions, EEHVs have been classified into eight subtypes that include EEHV1A, EEHV1B, and EEHV2-7, for which the clinical characteristics differ among EEHV subtypes [2,5,6,7,8,9]. EEHV1A, 1B, 4, and 5 mainly infect Asian elephants, while EEHV2, 3, 6, and 7 infect African elephants [2,7]. Currently, in vitro isolation of EEHV through the use of various cells has been tested intensively, but the tests have remained unsuccessful in recognizing EEHV as one of the most challenging viruses affecting elephant populations [10,11]. Presently, various diagnostic methods for EEHV detection have been developed and implemented in diagnostic and therapeutic approaches. These methods involve detection of the viral genome, detection of viral proteins, or detection of antibodies against EEHV infection. Viral genome detection includes PCR, qPCR, loop-mediated isothermal amplification (LAMP), and in situ hybridization [12,13,14,15,16,17]. Viral protein and viral particle detection involves histopathology, Western immunoblotting, immunohistochemistry, immunofluorescence, and electron microscopy [2,3,4,18,19], while anti-EEHV antibody detection involves enzyme-linked immunosorbent assay (ELISA) and the luciferase immunoprecipitation system (LIPS) [20,21,22,23,24].

Generally, the determination of viral infections using antibody detection has become preferable over other diagnostic tools because it is considered to be more cost effective, simple, less laborious, and has the capacity to screen a large number of samples. To date, most of the antibody tests developed for EEHV detection have been based on EEHV structural proteins (SP), namely, gB and gH/gL [21,22,23]. The presence of antibodies against viral SPs cannot be used to determine active viral replication in the host, since the presence of these antigens can be a sequence of either viral replication or degradation of the virus, during phagocytosis of antigen-presenting cells (APCs) [25,26]. Moreover, despite the fact that the PCR test has been widely used to determine active infection and shedding of EEHV through blood, feces, saliva, or trunk wash [2,16,17,27], it is highly costly when applied to large populations. This would likely make it inappropriate for use in a number of countries. Thus, the search for alternative diagnostic tools that are cost effective and can be used on large elephant populations is of significant interest. The present study aimed to develop and evaluate an in-house ELISA for the detection of active EEHV infection in Asian elephants.

## 2. Materials and Methods

### 2.1. Sample Collection and Ethical Statement

A total of 175 elephant serum samples were included in this study. Elephant sera were obtained from the auricular ear veins or elephant carcasses, and were segregated from whole blood clots by centrifugation. Sera samples were then stored at −20 °C until further analyzed. The serum samples were categorized into 4 groups, according to the age range of the elephants and their history of exposure to EEHV. The categorization of these sera samples was then used to establish a database of positive/negative EEHV-PCR results (Table 1).

Specimens included seven EEHV-PCR positive serum samples (group D) that were collected in Chiang Mai, Thailand, between the years 2017 and 2020, while 168 EEHV-PCR negative samples were obtained from Chiang Mai and Lampang provinces, Thailand, during the course of the Elephant Health Monitoring Program from 2017 to 2021. Relevant animal protocols were conducted according to the *Guide for the Care and Use of Laboratory Animals* (National Institute of Animal Health, WA), and were approved by the Faculty of Veterinary Medicine, Chiang Mai University Animal Care and Use Committee (License No. S22/2563).

### 2.2. Polymerase Chain Reaction (PCR)

DNA was extracted from elephant blood samples using a DNA extraction kit (Machery-Nagel GmbH, Dauren, Germany), according to the manufacturer’s instructions. Specific primers, namely, pan-polymerase and terminase-specific primers, were used in PCR analysis for the screening of EEHV, as previously described [2,18,19]. Briefly, samples were screened for EEHV using PAN-EEHV POL primers 6710/6711 [10]. Then, the positive samples were confirmed for the EEHV subtype using terminase-specific primers for EEHV1 [2] and terminase-specific primers for EEHV3/4, as previously described [6] (Appendix A). The specific EEHV subtypes were further determined by sequencing, as has also been described previously [18,19].

### 2.3. In-House Indirect ELISA

An in-house indirect ELISA test was conducted according to a previously published protocol [28], with minor modifications. Briefly, EEHV-DNApol proteins obtained from the genotype EEHV1A, 31.2 kDa of DNAPolF2E1 recombinant protein (expressed from *Escherichia coli* in pET32b+ vector) [19], were coated on a Maxibinding Immunoplate^®^ (SPL Life Science, Korea) with a carbonate bi-carbonate coating buffer at a pH of 9.6 at 100 µL/well (final concentration 15 µg/mL). The proteins were then incubated overnight at 4 °C. Thereafter, plates were further incubated with 100 µL/well of blocking buffer (5% *w*/*v* skim milk in 0.01M phosphate buffer saline, PBS pH 7.4) at 37 °C for one hour. Subsequently, plates were washed 5 times with washing buffer (0.01M PBS + 0.05% *v*/*v* Tween-20) at 200 µL/well. Elephant sera were diluted 1:200 times in blocking buffer using vortex generation. Accordingly, 100 µL/well was added and the specimens were then incubated at 37 °C for 2 h. After being washed, HRP-conjugated recombinant protein G (1 mg/mL; Sigma-Aldrich, Burlington, MA, USA) was added and the specimens were further incubated at 37 °C for one hour. Plates were then washed again and color development was achieved by the addition of 3,3′,5,5′-tetramethylbenzidine (TMB; SeraCare Life Sciences Inc., Milford, MA, USA) for 5 min in the dark. Finally, reactions were stopped by adding 50 μL/well of 2 M sulfuric acid. A bright yellow color appeared at an absorbance value of 450 nm, and the specimens were evaluated with an ELISA plate reader (AccuReader, Metertech, Taipei, Taiwan, ROC).

### 2.4. Optimization of Antigens and Serum Concentration

To optimize relevant dilutions of antigen and serum concentrations, checkerboard titration was performed, according to the method previously described [29]. Briefly, antigens were titrated two-fold, from 30 µg/mL to 3.75 µg/mL, on ELISA plates. HRP-conjugated recombinant protein G was then titrated from 1:1000 to 1:5000, while elephant sera were titrated at a ratio of 1:200. Optimal conditions were then derived from the resulting dilutions, for which optimal ratios for positive and negative samples of optical density (OD) were used for each of the ELISA tests.

### 2.5. Determination of Cut-Off Values

To determine the cut-off values of EEHV-DNApol ELISA, 14 serum samples from the group that had no previous history of EEHV infection and that had tested negative by PCR (group A) were used. Cut-off values were established by determining the average values of the negative samples +3 standard deviation (SD) values, as has been previously described [28]. Serum samples displaying an OD_450_ value greater than or equal to the cut-off value were considered EEHV-seropositive.

### 2.6. Sensitivity Estimation

To assess the sensitivity of the ELISA test, serum samples were evaluated and the results were compared to those of a PCR test. Since there is no gold standard for EEHV detection, sensitivity values were estimated by latent class analysis using a Bayesian model, as has been previously described [30]. An in-house indirect ELISA test was developed to detect anti-EEHV antibodies; therefore, the results were considered conditionally dependent upon the results of the prior analysis. On the other hand, the PCR test aimed to detect the specific subsequence of the viral genome; hence, the results were considered to be conditionally independent of the ELISA test [31,32]. Prior information on the test performance results and the prevalence of EEHV was introduced into the analysis via probability distributions (prior distributions). There is a lack of available information on the performance of the ELISA test and PCR results, in terms of the diagnosis of EEHV infection; thus, the prior values that had been derived from three expert opinions served to represent a model for beta distribution. All analyses were implemented in JAGS via ‘rjags’ and ‘R2jags’ packages using R 4.1.0 software [33,34,35]. Posterior distributions were computed after 100,000 iterations of the models, with the first 10,000 iterations being discarded as part of the burn-in phase. Sensitivity value of EEHV-DNApol ELISA was then defined.

### 2.7. Cross-Reactivity of Indirect ELISA Results among Elephant Sera and Those of Other Animals

To determine the specificity of the ELISA test, sera obtained from different animal species were evaluated. The 2-fold diluted antigen (15 to 0.234 µg/mL) and sera collected were from different animal species, including fallow deer (*Dama dama*), smooth-coated otter (*Lutrogale perspicillata*), equine (*Equus caballus*), cattle (*Bos indicus*), macropus (*Macropus* sp.), eland (*Taurotragus oryx*), and the Malayan sun bear (*Helarctos malayanus*). The samples were used to identify the antigen specificity to elephants using indirect ELISA. The sera were obtained from the Sample Bank of the Veterinary Diagnostic Laboratory, Faculty of Veterinary Medicine, Chiang Mai University, Thailand. The percentage of binding (% binding) for the sera-to-antigen ratio was compared to that of the elephant at 100% of the 15 µg/mL antigen concentration, as has been previously described [36].

## 3. Results

### 3.1. Optimization of EEHV-DNApol ELISA

As was determined by checkerboard titration, the optimal antigen concentration of EEHV-DNApol for the ELISA test was found to be 15 µg/mL (Appendix A). The optimal serum sample dilution was shown to be 1:200, while the optimal dilution of the HRP-conjugated recombinant protein G (1 mg/mL) was determined to be 1:1500 (Appendix A).

### 3.2. Determination of Sensitivity and Specificity Values of ELISA

Posterior estimations of the sensitivity values for the EEHV-DNApol ELISA test were lower than for all of the prior values, with a median score of 77.9% (PPI = 52.5–95%), while the estimated specificity value was considered to be similar to the prior values, with a median score of 87.7% (PPI = 82.5–91.9%) (Table 2). Meanwhile, the estimated sensitivity values of the PCR test were lower than the prior values, with a median score of 88.9% (95% PPI = 68.3–98.3%), while the specificity value was determined to be 96.6% (95% PPI = 93–99.1%). Notably, the posterior estimation for the prevalence of the disease when tested by the EEHV-DNApol ELISA was considerably lower than the prior score, with a median value of 5.5% (95% PPI = 2.4–20.7%) (Table 2).

In terms of the convergent test results, the model exhibited proper convergence, while autocorrelation was eliminated after omitting the first 10,000 iterations. No major changes (change in median or 95% probability percentiles >20%) in the posterior estimates of the specificity values of the tests were found when non-informative distributions were employed as the prior values for any of the parameters. The results of the sensitivity analysis were interpreted as evidence of the model’s level of robustness. Nevertheless, changes were observed in the posterior estimates for the sensitivity values of the EEHV-DNApol ELISA test results and the PCR results, while the degree of prevalence was observed at a lower level of estimated posterior sensitivity.

### 3.3. Determination of the Cut-Off Value and Detection of EEHV Antibodies

Based on the ELISA data established from EEHV-negative animals, the cut-off values for EEHV-DNApol ELISA were determined from a cut-off point of 2.144. The developed EEHV-DNApol ELISA was then applied to 175 samples of elephant sera, with various groups of EEHV status. Samples from elephant groups A, B, C, and D revealed average OD_450_ values of 1.038 ± 0.368, 0.996 ± 0.301, 1.340 ± 0.639, and 0.556 ± 0.714, respectively (Appendix A).

The lowest levels of the antibody against EEHV were observed in elephants that had died due to EEHV-HD (group D), when compared to other groups with average OD_450_ values recorded at 0.556 ± 0.714 (Appendix A). It should be noted that only sera obtained from the elephant group of unknown history with regard to EEHV infection (group C) indicated seropositivity for EEHV-DNApol ELISA at 14% (21/150), while the sera were found to be negative by the PCR test (Table 3). The EEHV seropositivity results of group C elephants were obtained from elephants that were less than 4 years old (9.52%; 2/21), elephants that were 4–10 years old (14.28%; 3/21), and elephants that were more than 10 years old (76.2%; 16/21).

### 3.4. Comparison of ELISA and PCR Test Results

The EEHV-DNApol ELISA antibody detection test and the viral genomic DNA test (PCR) were administered to test 175 samples obtained from the elephants of groups A, B, C, and D. The results revealed that 84% (147/175) of the samples tested negative when both tests were administered, while no elephants tested positive when both tests were applied. It should be noted that while some samples tested negative by PCR, 12% (21/175) of samples were found to be positive with the use of EEHV-DNApol ELISA (Table 4). Those elephants were in a group of unknown history for EEHV infection (group C). Meanwhile, samples that tested positive by PCR (7/175) were obtained from a group of elephants that had died due to acute EEHV-HD.

### 3.5. Cross-Reactivity of EEHV-DNApol ELISA Results

To examine the cross-reactivity of EEHV-DNApol ELISA results with other animals, sera obtained from fallow deer, smooth-coated otter, equine, cattle, Macropus, eland, and Malayan sun bear were tested. The results indicated that the developed ELISA test specifically reacted to the elephant sera, followed by sera from equine and fallow deer, respectively (Figure 1).

## 4. Discussion

During the past decade, one of the most highly fatal diseases in young Asian elephants has been attributed to infection with EEHV. As the infection of EEHV in adult Asian elephants usually manifests as asymptomatic, coupled with being non-clinically ill, these animals are believed to be EEHV shedders in elephant herds [13,37]. Thus, affordable diagnostic tools for the screening and identification of active EEHV infection and shedders among a large number of elephants would be highly beneficial. Our previous study demonstrated that antibodies against the EEHV-DNApol nonstructural protein could be used to determine EEHV replication in Asian elephants in vivo [19]. In the present study, we further extrapolated the results of our previous work by developing an in-house indirect ELISA test for the detection of EEHV-DNApol antibodies. The results of the EEHV-DNApol ELISA test were then evaluated and compared to the results of the PCR test using the Bayesian statistical model.

In agreement with the findings of previous reports, we have demonstrated that adult Asian elephants likely serve as EEHV shedders within elephant herds [17,37,38,39,40]. A serological study for EEHV detection using ELISA tests has been implemented in elephants in European zoos, as well as those living in captivity in North America and Thailand, using EEHV structural proteins such as gB and gH/gL [21,23,24]. In contrast to the results of previous reports, however, the seropositivity results for the EEHV-DNApol ELISA test in this study were lower than those of previous reports, indicating 42–44% seropositivity for EEHV-gB ELISA in Thai captive elephants [22,24]. Moreover, a serosurvey of EEHV infection using the EEHV-gH/gL ELISA revealed that more than 97% of elephants from European zoos were determined to be EEHV seropositive [23]. The discrepancy of our results with those of previous reports could be explained by the fact that different EEHV antigens were used, since most of those studies employed EEHV structural proteins [21,23,24]. In light of our current understanding, it remains to be determined whether antibodies against structural proteins could be indicative of active or latent EEHV infection, as it is known that latent and reactivated herpesvirus infection could elicit an antibody response against both structural and nonstructural proteins [41,42]. Nevertheless, the antibody responses against nonstructural proteins could only be determined by active infection or reactivation of the herpesvirus, which would allow us to identify EEHV shedders in elephant herds using EEHV-DNApol ELISA. In addition, our findings agree with the results of previous studies that reported that EEHV-HD-related sera showed low or non-detectable antibody levels, further indicating EEHV primary infection in young Asian elephants [20]. However, it should be noted that despite the fact that these animals succumbed to EEHV-HD, the elephant D1 group exhibited high antibody levels, whereas all elephants in group D were found to be negative by the EEHV-DNApol ELISA test. This determination may be attributed to the fact that the elephants in the D1 group could produce the antibody against EEHV before death, while elephants in the D2-D7 groups succumbed to the virus after just a few days, prior to generation of the EEHV-primed T cell [43].

In the present study, the Bayesian analysis model was used to determine the degrees of sensitivity and specificity of the ELISA values in comparison to those of the PCR test because there is currently no gold standard for EEHV detection. However, there are some factors that may have affected the cut-off calculations for the selected serum samples. Firstly, the sudden death of EEHV-HD elephants (group D) may have occurred due to EEHV primary infection, which would have been associated with low or non-detectable EEHV-specific antibody titers. Secondly, elephants in the EEHV-negative group (group A) may have been exposed to EEHV, but did not show clinical signs. This was evident despite the fact that the tests were unable to detect viral DNA at the time of testing. Thus, the true negative sera of EEHV infection in Asian elephants remains to be determined and needs to be further evaluated. Moreover, there are several limitations of our currently developed ELISA test. Firstly, it is impossible to distinguish between different subtypes of EEHVs. Secondly, the sensitivity of the ELISA test remains low in detecting acute EEHV infection, since elephants may require more than one to two weeks to develop antibodies against the nonstructural protein of the virus. However, since our developed ELISA test is inexpensive and achievable for the screening of a large number of elephants, it may be useful for routine monitoring of elephant herds. In summary, the present study has demonstrated that an in-house indirect EEHV-DNApol ELISA test could be used as an effective alternative diagnostic tool for the detection of active EEHV infection and shedders in Asian elephants.

## 5. Conclusions

The present study has indicated that asymptomatic, active EEHV-infected elephants could likely serve as a source of EEHV shedding within elephant herds. The developed EEHV-DNApol NSPs-based ELISA test employed in the present study may be of use for routine monitoring and identification of EEHV shedders in elephant herds, and could be an alternative diagnostic tool for EEHV detection in Asian elephants.

## Figures and Tables

**Figure 1 animals-12-01747-f001:**
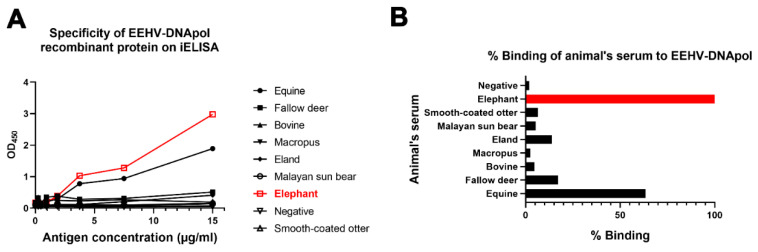
Cross-reactivity of EEHV-DNApol ELISA results with animal sera. Elephant serum (red) indicated 100% specificity with the EEHV-DNApol recombinant protein, followed by equine and fallow deer at 63.5% and 17.2%, respectively (**A**,**B**). Other sera indicated low %binding (less than 20%) to EEHV-DNApol antigens.

**Table 1 animals-12-01747-t001:** Categorization of elephant sera used to develop an in-house nonstructural protein ELISA.

Group	PCR	History of EEHV Infection	Number of Samples	Age Range (Years Old)
A	Negative	No previous clinical signs of EEHV infection	14	2–11
B	Negative	Had shown EEHV clinical signs and tested positive by PCR in 2017 or 2018 then recovery	4	6–8
C	Negative	Unknown	150	2–80
D	Positive	Sudden death from EEHV-HD (within 1–7 days after showing clinical signs)	7	2–7
Total			175	2–80

EEHV-HD: elephant endotheliotropic herpesvirus hemorrhagic disease.

**Table 2 animals-12-01747-t002:** Prior values and posterior estimation for sensitivity and specificity of the EEHV-DNApol ELISA test, the PCR test, and the degree of prevalence of the disease (%).

Diagnostic Test	Parameters	Prior Value		Posterior Estimates	
		Mode (%)	95% CI ^a^	Median (%)	95% PPI ^b^
EEHV-DNApol ELISA	Sensitivity	90	>50.0%	77.9	52.5–95
	Specificity	85	>50.0%	87.7	82.5–91.9
PCR	Sensitivity	95	>80.0%	88.9	68.3–98.3
	Specificity	95	>80.0%	96.6	93–99.1
Prevalence		40	<60.0%	5.5	2.4–20.7

^a^ 95% CI: credibility interval ^b^ 95% PPI: posterior probability interval.

**Table 3 animals-12-01747-t003:** Detection of EEHV antibodies using EEHV-DNApol ELISA on various groups of elephant sera.

Group	ELISA		PCR	
	Positive	Negative	Positive	Negative
A	0% (0/14)	100% (14/14)	0% (0/14)	100% (14/14)
B	0% (0/4)	100% (4/4)	0% (0/4)	100% (4/4)
C	14% (21/150)	86% (129/150)	0% (0/150)	100% (150/150)
D	0% (0/7)	100% (7/7)	100% (7/7)	0% (0/7)

**Table 4 animals-12-01747-t004:** Combined results for the detection of the antibody (ELISA) and viral genomic DNA (PCR) in 175 samples.

EEHV-DNApol ELISA	PCR	Number
−	−	147
+	−	21
−	+	7
+	+	0
Total		175

## Data Availability

All datasets generated or analyzed during this study are included in the published article.

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
