# Peer review of "Development of Nonstructural Protein-Based Indirect ELISA to Identify Elephant Endotheliotropic Herpesvirus (EEHV) Infection in Asian Elephants (Elephas maximus)"

_animals, 2022, doi:10.3390/ani12141747_

Round 1

Reviewer 1 Report

The author have clarifiied the questions and revised the manuscript accodingly. But the writing of the manuscript remains to be improved. 

Author Response

Reviewer #1:

The author have clarified the questions and revised the manuscript accodingly. But the writing of the manuscript remains to be improved. 

A: We appreciate for your kind comment. As suggested, we have now slightly edited the manuscript, and the manuscript is already language proved by the native English speaker (please see Manuscript).

Reviewer 2 Report

Major comment:

For me, the biggest weakness of the work is the lack of statistical analysis of the results.

Please include data and statistical analysis and binomial confidence intervals of 95%!

In the abstract, I would like to see the overall incidence of EEHV detected by PCR!  example % (n=?/175).

After that, you must present the overall seroprevalence %  (n=?/175) and after that, you can give the sensitivity and specificity of your test.

The quality of figure 1 is poor!

Author Response

Reviewer #2:

  1. For me, the biggest weakness of the work is the lack of statistical analysis of the results. Please include data and statistical analysis and binomial confidence intervals of 95%!

A: We appreciate for your kind comment. As suggested, we have now included the posterior probablity estimation (PPI) of 95% of the sensitivity and specifcity for the EEHV-DNApol ELISA to the Abstract (please see Abstract, page 2, line 44).

  1. In the abstract, I would like to see the overall incidence of EEHV detected by PCR!  example % (n=?/175). After that, you must present the overall seroprevalence %  (n=?/175) and after that, you can give the sensitivity and specificity of your test.

A: We appreciate for this comment. As we mentioned the results of PCR test in Table 4 of the Result section and discussed this point in the Discussion section. It was 4% (7/175) of the PCR positive samples which were obtained from animals that succumbed from the acute EEHV-HD cases and all of these samples were shown to be negative by the EEHV-DNApol ELISA (please see Table 4). These results could be due to the fact that those elephants may have been exposed to the virus after just a few days prior to generation of the EEHV-primed t-cell. Thus, the antibody level of animals in this group was low or non-detectable when using the ELISA test (please see Results, page 7, line 245-253, Table 4, and Discussion, page 9, line 304-310). On the other hand, 12% (21/175) of the ELISA positive samples were observed in elephants group of unknown history, which these samples were shown to be negative by the PCR test (please see Results, page 7, line 245-253, Table 4). These results suggested that the animals in this group may had been exposed with active EEHV infection. Subsequently, the virus may then clear from the elephant blood circulation by immune responses, and thus, only antibodies against viral non-structural proteins were detected. However, since the present study was aimed to develop the test for screening of asymptomatic, active EEHV-infected elephants which could likely be serve as shedder in the elephant herds, we then emphasized only the seroprevalence of animals in this group, instead of both with the PCR results, in the Abstract (please see Abstract, page 2, line 45-46). Results of overall PCR tests and overall seroprevalence in this study have already mentioned in the Result section (please see Results, page 7, line 232-253, Table 4).

  1. The quality of figure 1 is poor!

A: We appreciate for this comment. The quality of Figure 1 has now been enhanced (please see Figure 1).

Reviewer 3 Report

The authors have adequately addressed my concerns in their revised manuscript.

I only have one minor comment that texts in Fig. 1 are still not sharp enough. Since there is only one Figure in this paper, I highly suggest providing a high-resolution publication-quality image.

Author Response

Reviewer#3

The authors have adequately addressed my concerns in their revised manuscript.

I only have one minor comment that texts in Fig. 1 are still not sharp enough. Since there is only one Figure in this paper, I highly suggest providing a high-resolution publication-quality image.

A: We appreciate for this comment. As also suggested by the Reviewer #2, the quality of Figure 1 is now improved (please see Figure 1).

Round 2

Reviewer 2 Report

On the other hand, 12% (21/175) of the ELISA positive samples were observed in elephants group of unknown history, which these samples were shown to be negative by the PCR test (please see Results, page 7, line 245-253, Table 4). These results suggested that the animals in this group may had been exposed with active EEHV infection. Subsequently, the virus may then clear from the elephant blood circulation by immune responses, and thus, only antibodies against viral non-structural proteins were detected. 

If the virus has been cleared from the blood and can not be detected by PCR we can not talk of active infection. I think that you have a problem with the sensitivity of the PCR reaction. 

Since western blot is more specific and confirmatory than ELISA I suggest to authors confirm their results from the ELISA test with Western blot.

Author Response

Response to the reviewer’s comments

(Submission ID: animals-1785138)

Reviewer #2:

  1. If the virus has been cleared from the blood and can not be detected by PCR we can not talk of active infection. I think that you have a problem with the sensitivity of the PCR reaction. 

Since western blot is more specific and confirmatory than ELISA I suggest to authors confirm their results from the ELISA test with Western blot.

A: We are grateful for this comment and do apologize for our obscure statement. As mentioned by the Reviewer, we agree that if the virus has been cleared from the blood circulation, one should consider when using the term active infection. As we have been mentioned in the previous version of our ”Response to the reviewer’s comment” and the manuscript, when the virus replicates in host cells, the non-structural proteins required for virus replication, such as EEHV DNA polymerase, may be detected and can be used to determine active infection of EEHV in elephants (please see version 1, Response to the reviewer’s comment). Therefore, host antibody responses against the EEHV DNA polymerase protein can be applied to use as an indicator for EEHV infection and replication (active infection), despite it may require few days to weeks after virus entering into host cells (Tizard I. 2016).

Thus, to clarify this point as first mentioned by the Reviewer, we now added the incidence of detection by PCR and the overall seroprevalence by the EEHV DNApol ELISA to the abstract, as suggested (please see Abstract, page 2, line 48-50).   

References

Tizard I. 2016. Veterinary immunology an introduction 10th edition. W.B. Saunders

Company. 401 pages

Round 3

Reviewer 2 Report

The results of the ELISA test are not convincing enough to claim that this test is suitable for detecting active infection. The original title of the manuscript was " Development of nonstructural protein-based indirect ELISA to identify elephant endotheliotropic herpesvirus (EEHV) infection in Asian elephants (Elephas maximus) " active was missing and my opinion is that this ELISA test cannot prove an active infection.

In Materials and Methods you lack a description of the PCR method, you have the citation of three different articles, which does not allow me to find out which method you used for PCR analysis. Please describe precisely the PCR methodology that you have applied and the respective primer pairs.

Author Response

Response to the reviewer’s comments

(Submission ID: animals-1785138)

Reviewer #2:

  1. 1. The results of the ELISA test are not convincing enough to claim that this test is suitable for detecting active infection. The original title of the manuscript was "Development of nonstructural protein-based indirect ELISA to identify elephant endotheliotropic herpesvirus (EEHV) infection in Asian elephants (Elephas maximus)" active was missing and my opinion is that this ELISA test cannot prove an active infection.

A: We are grateful for this comment. As suggested, we changed the title to the original version as ”Development of nonstructural protein-based indirect ELISA to identify elephant endotheliotropic herpesvirus (EEHV) infection in Asian elephants (Elephas maximus)", (please see Title, page 1).  

  1. In Materials and Methods you lack a description of the PCR method, you have the citation of three different articles, which does not allow me to find out which method you used for PCR analysis. Please describe precisely the PCR methodology that you have applied and the respective primer pairs.

A: We appreciated for this comment. As suggested, the detail of primers used in this study are now rewritten and cited in Materials and Methods and Supplementary 1 Table (please see Materials and Methods, page 4, line 120-129 and Supplementary data). In addition, the S2 and S3 Tables have been updated accordingly in the Manuscript (please see Manuscript).

This manuscript is a resubmission of an earlier submission. The following is a list of the peer review reports and author responses from that submission.

Round 1

Reviewer 1 Report

Overview and general recommendation:

Guntawang and colleagues presented the establishment of an in-house indirect ELISA, which can be used for routine screening of anti-EEHV antibodies in the serum samples of elephants. The authors demonstrated the high sensitivity (77.9%) and specificity (87.7%) of this novel ELISA based on partial EEHV DNApol NSPs. Considering the significant mortality of EEHV infections in Asian elephant calves, the established ELISA in this study may be beneficial for the diagnosis and control of EEHV infections. However, some points in the manuscript are not explicitly clarified, and the manuscript needs to be extensively improved. I have the following comments for the authors’ consideration.

  1. My major concern is that the sensitivity and specificity of the unique ELISA established in this study are calculated based on the computational analyses. According to the M&M 2.6., if I understand correctly, the estimated sensitivity and specificity are highly dependent on each number of different groups, which is weird. The reviewer disagrees the sensitivity and specificity of an ELISA can be precisely determined from a Bayesian model.
  2. Moreover, regarding the specificity (M&M 2.7. and Results 3.5.), the authors used different animal serum samples to prove that there is no cross-reactivity. However, they should also determine that the ELISA is specific to anti-EEHV antibodies but not anti-other viruses’ antibodies in elephant serum samples.
  3. The authors compared the positivity of EEHV infection using ELISA and PCR, which seems improper. Why don’t the authors test these 175 serum samples of Asian elephants with the EEHV SP-base ELISA? I noticed that one co-author in this study detected the seroprevalence of EEHV in Asian elephants using the EEHV SP-base ELISA previously (see References 25 and 42). A comparative analysis of EEHV DNApol-based and SP-base ELISA would be advantageous.
  4. In the Introduction section, the authors mentioned that there are currently 8 subtypes of EEHVs; please specify the distinct properties of these subtypes. Notably, it would be helpful if the authors provided the EEHV subtypes of 7 EEHV-PCR-positive samples in this study (Materials and Methods 2.2.).
  5. In the Introduction section, paragraph 2, lines 5-8. The statement is confusing to me. Did the authors want to underlie that the DNApol-based ELISA is suitable for the detection of both acute and past EEHV infection; in comparison, the SP-based ELISA is only for the detection of past EEHV infection? If so, the detection rates from DNApol-based ELISA should be theoretically higher than SP-based ELISA. However, the actual seroprevalence derived from these two different ELISA is conflicting in the Discussion section. Is there any reasonable explanation? Or the DNApol-based ELISA is applicable for acute EEHV infection?
  6. In the Materials and Methods section 2.3, the description of the development of this unique EEHV-DNApol ELISA is very limited. For example, what is the specific targeting region of the EEHV-DNApol? How is this EEHV partial NS protein produced? Since this study focuses on establishing a diagnostic method, a mere citation of previous work (reference 19) is apparently not enough.
  7. In the Materials and Methods section 2.6, the title is misleading since the specificity is not outlined in this part but in the next part.
  8. In the discussion section, paragraph 2, if the EEHV-DNApol ELISA can detect acute EEHV infection, how to explain all group D PCR-positive samples displayed ELISA-negative results (Table 3)? Additionally, what do the authors mean by the elephant D1 to D7? These terms should be clearly defined in the M&M and Results section.
  9. In the first sentence of the figure legend of Figure 1, it is stated that the elephant serum showed 100% specificity with the EEHV-DNApol recombinant protein. In contrast, in the 3.2 of the Results section, the specificity is 87.7%. This description seems confusing to me. In addition, Figure 1 is too blurred to read, and please increase the resolution of this figure.
  10. I doubt that the authors properly used a citation manager for the references because I found that references 2 and 20 are identical, and references 25 and 42 are the same. Please check the references carefully.
  11. There are numerous grammatical errors in this manuscript. Please see the minor points in detail.

Minor points:

  1. Abstract, line 1: change “Abatract” to “Abstract”.
  2. Abstract, line 2: change “diseases” to “disease”.
  3. Abstract, line 7 and Introduction, line 17: change “enzyme linked” to “enzyme-linked”.
  4. Introduction, line 4: change “hemorrhaging” to “hemorrhage”.
  5. Introduction, line 9: change “virus” to “viruses”.
  6. Introduction, paragraph 2, lines 2 and 12: change “cost effective” to “cost-effective”.
  7. Introduction, paragraph 2, line 2: change “antigen presenting cells” to “antigen-presenting cells”.
  8. Introduction, last sentence: delete “assay”.
  9. Materials and Methods, 2.3, line 14: change “the adding of” to “adding”.
  10. Materials and Methods, 2.4, line 4: add “from” after “titrated”.
  11. Materials and Methods, 2.7, line 2: add “were” after “collected”.
  12. Results, 3.3, line 3: change “were” to “was”.
  13. Results, 3.3, paragraph 2, line 2: please define “EEHV-HD”.
  14. Results, 3.3, line 7: change “those” to “elephants”.
  15. Results, 3.3, lines 7-8: change “years-old” to “years old”.
  16. Results, 3.3, line 8: delete the first “were”.
  17. Discussion, paragraph 2, line 12: change “employing” to “employed”.
  18. Discussion, paragraph 2, line 12: change “employing” to “employed”.
  19. Discussion, paragraph 2, line 12: add “as” before “active”.
  20. Discussion, paragraph 3, line 15: change “develop” to “developed” and “screen” to “screening”.
  21. Discussion, paragraph 3, line 15: change “in house” to “in-house”.

Reviewer 2 Report

Comments to the Author
Authors: Thunyamas Guntawang, Tidaratt Sittisak, Pallop Tankaew, Chatchote
Thitaram, Varangkana Langkapin, Taweepoke Angkawanish, Tawatchai Singhla,
Nattawooti Sthitmatee, Wei-Li Hsu, Roongroje Thanawongnuwech, Kidsadagon
Pringproa *

Title: Development of nonstructural protein-based indirect ELISA to identify
elephant endotheliotropic herpesvirus (EEHV) infection in Asian elephants
(Elephas maximus)

In this manuscript, Thunyamas Guntawang and co-authors address key issues related to identify elephant endotheliotropic herpesvirus (EEHV) infection in Asian elephants and developed an ELISA diagnostic test.

Major comment:

In introduction I would like to see more information about EEHV.

Please, enter the disease caused by the virus - Elephant endotheliotropic herpesvirus-hemorrhagic disease (EEHV-HD)

Please, include the subfamily Betaherpesvirinae of EEHV.

Please, give more information about the genome organization and the ORF and the proteins that they code. Explain the role a EEHV-DNA polymerase (DNApol) nonstructural proteins (NSPs), explain why you choose the  DNApol as a detection Ag?

Materials and Methods

Include Table 1 directly after it citation.

In-house indirect ELISA – write the concentration of the coating protein EEHV-DNApol.

In materials and methods nowhere did I see the origin of the coating protein EEHV-DNApol! Please explain the origin of  the EEHV-DNApol.

The quality of Figure 1 is poor, please fix it.

Discution

nonclinically ill symptoms - there is no such term

In addition, our findings agree with the results of previous studies which had reported that EEHV-HD related sera showed low or non-detectable antibody levels further indicating EEHV primary infection in young Asian elephants. – In this sentence, it is not clear that you are talking about antibodies against DNApol. Please specified your sentence!

The ELISA test you created has many shortcomings and cannot be used as a reliable method for diagnosing of EEHV-HD, but the idea of looking for antibodies against DNA pol is very good.

Reviewer 3 Report

This manuscript developed a ELISA method to identify EEHV infection in Asia elephant using EEHV non-structural proteins. The  authors compared the sensitivity and speicificity and determined the cut off value that can be tested.  This method are benificial to the EEHV infection but it is not a novel method for the EEHV infection because the strutral protein of EEHV has been used for the identification of EEHV infection. There are some questions in the manuscript. 

  1. There is no introdction about why the non structural protein of EEHV used in this study.
  2. There is no desciption about the fragment of the non structural protein of EEHV used in this study. 
  3. In 2.6 Sentivity and Specificity estimation, majority of this desciption did not belong to the Materials and Methods.
  4. Results 3.1: why the optimal dilution of HRP is 1:1500 not 1:1000 beacuse at this interaction the OD450 vaule is maximal.
  5. S2 Table: The SD of group D is larger than its average. In addition,  what is the effect of the age distribution of group C in the EEHV antibody detection?
  6.  By comparing the ELISA  and PCR results, what is conclusion ?The ELISA is superior to PCR or  something? This is not discussed.
